# Vasoactive Intestinal Peptide Receptor, CRTH2, Antagonist Treatment Improves Eosinophil and Mast Cell-Mediated Esophageal Remodeling and Motility Dysfunction in Eosinophilic Esophagitis

**DOI:** 10.3390/cells13040295

**Published:** 2024-02-06

**Authors:** Chandra Sekhar Yadavalli, Sathisha Upparahalli Venkateshaiah, Alok K. Verma, Chandrasekhar Kathera, Pearce S. Duncan, Michael Vaezi, Richard J. Paul, Anil Mishra

**Affiliations:** 1Department of Medicine, Section of Pulmonary Diseases, Tulane Eosinophilic Disorder Center, School of Medicine, Tulane University, New Orleans, LA 70112, USA; cyadavalli@tulane.edu (C.S.Y.); sathisha114@gmail.com (S.U.V.); ckathera@tulane.edu (C.K.); 2Division of Gastroenterology, Cincinnati Childrens Medical Center, Cincinnati, OH 45229, USA; alok.verma@cchmc.org; 3Division of Gastroenterology, School of Medicine, Tulane University, New Orleans, LA 70118, USA; dpearce3@tulane.edu; 4Division of Gastroenterology, Vanderbilt University Medical Center, Nashville, TN 37232, USA; michael.vaezi@vumc.org; 5Division of Physiology, Cincinnati University, Cincinnati, OH 45220, USA; richard.paul@uc.edu

**Keywords:** CRTH2, eosinophils, esophagus, EoE, mast cells, motility

## Abstract

Background and Aims: Ultrasonography has shown that eosinophils accumulate in each segment of the esophageal mucosa in human EoE, ultimately promoting esophageal motility dysfunction; however, no mechanistic evidence explains how or why this accumulation occurs. Methods: Quantitative PCR, ELISA, flow cytometry, immunostaining, and immunofluorescence analyses were performed using antibodies specific to the related antigens and receptors. Results: In deep esophageal biopsies of EoE patients, eosinophils and mast cells accumulate adjacent to nerve cell-derived VIP in each esophageal segment. *qRT*-PCR analysis revealed five- to sixfold increases in expression levels of *VIP*, *CRTH2*, and *VAPC2* receptors and proteins in human blood- and tissue-accumulated eosinophils and mast cells. We also observed a significant correlation between mRNA *CRTH2* levels and eosinophil- and nerve cell-derived *VIP*s in human EoE (*p* < 0.05). We provide evidence that eosinophil and mast cell deficiency following CRTH2 antagonist treatment improves motility dysfunction in a chronic DOX-inducible CC10-IL-13 murine model of experimental EoE. Conclusions: CRTH2 antagonist treatment is a novel therapeutic strategy for inflammatory cell-induced esophageal motility dysfunction in IL-13-induced chronic experimental EoE.

## 1. Introduction

Esophageal eosinophilia is commonly observed in diverse gastrointestinal problems, including eosinophilic esophagitis (EoE) [1,2,3,4,5,6,7]. EoE is a global health concern, particularly for pediatric patients [8,9,10,11,12,13]. The characteristic endoscopic features observed in EoE patients include corrugated esophagus, esophageal remodeling, strictures, and motility dysfunction [8,14,15,16,17,18]. Halting or reversing progressive fibrostenosis can be clinically challenging, leaving untreated and treatment-resistant patients at risk of esophageal rigidity and food impactions. Many patients are unresponsive to current therapies; in the observation phase of a randomized, double-blind, double-dummy trial, the disease overwhelmingly reoccurred after an initial successful treatment [19]. EoE is closely linked to gastroesophageal reflux disease (GERD) but is distinguished by the lack of response to two weeks of proton pump inhibitor (PPI) treatment [19]. Evidence indicates that food allergens have an important role in promoting EoE pathogenesis [20,21,22]. For many years, most basic and clinical researchers and health care providers believed that EoE occurs in response to Th2 cell activation following allergen exposure [23,24,25]. In recent years, major advances have been made in understanding the genetic associations in patients with EoE [23,24,25]. Several reports indicate that inflammation caused by the accumulation of eosinophils and mast cells promotes esophageal fibrosis, shortening, stricture, and motility dysfunction in human EoE [26,27]. Previous studies revealed the potential role of nerve cell-derived vasoactive intestinal peptide (VIP) in the accumulation, activation, and degranulation of eosinophils [28,29,30] and mast cells in tissues [31]. Cell-derived mediator VIP and eotaxin-3 are also reportedly linked to eosinophil accumulation in the esophagus [28,29,30,32]. Mepolizumab (anti-IL-5) and anti-IL-13 neutralization therapy have been shown to reduce esophageal eosinophilia [33,34], but neither anti-IL-5 nor anti-IL-13 therapy can completely restrict EoE pathogenesis in humans. The FDA recently approved a phase 3 randomized, double-blind, placebo-controlled clinical trial for Dupixent (dupilumab, an anti-IL4Rα treatment); however, the efficacy and safety of a 300 mg dose of Dupixent for patients aged 12 and older has only been proven on a limited number of patients, providing 69% and 64% reduction in EoE symptoms from baseline compared to 32% and 41% with placebo. The endpoint showed only 30% EoE improvement by Dupixent compared to placebo [35], but its efficacy at improving esophageal rigidity, remodeling, stricture, and motility dysfunction in human EoE is unclear.

The current study will test the hypothesis that nerve cell-derived mediator VIP may have a role in the accumulation of eosinophils in esophageal epithelial mucosa and beyond the epithelium mucosa, including muscularis mucosa, that induce motility dysfunction. Our findings provide an additional insight into eosinophil and mast cell accumulation. We examined novel clinically relevant hypotheses and employed new approaches and technologies to determine the consequences of VIP-induced inflammation on the subsequent esophageal pathophysiology. We provide evidence that the interaction of nerve cell-derived VIP with receptors present on eosinophils expressing CRTH2 and mast cells expressing VIP receptors, CRTH2 and VAPC2, promote migration and accumulation to nerve cells in all esophageal segments. Since the VIP receptor CRTH2 is expressed on both eosinophils and mast cells, blocking it may prevent both cell types from accumulating nearby VIP-expressing nerve cells. The logical significant findings of the study show that migration and interaction between nerve cell-derived VIP expression and VIP receptors present on eosinophils and mast cells interaction is critical in the accumulation, activation, and degranulation of eosinophils and mast cells in each layer of the esophageal mucosa, including muscularis mucosa, that control esophageal motility. We expect that the current findings will establish nerve cell-derived factors, and their receptors may be novel therapeutic targets to improve human EoE pathophysiological functional abnormalities, including esophageal motility.

## 2. Materials and Methods

### 2.1. Clinical Characteristics of EoE and Non-EoE Patients

Following an IRB-approved protocol, normal and EoE esophageal biopsy samples were formalin-fixed and paraffin-embedded. Non-EoE participants and patients with dysphagia were recruited as comparative controls without considering age, atopic status, or sex. Diagnosis was based on the eosinophil count per HPF (×400). Patients with EoE had ≥15 esophageal eosinophils/HPF, while non-EoE subjects had 0–2 esophageal eosinophils/HPF and no basal layer growth. Patients who exhibited EoE symptoms but had normal esophageal endoscopic and microscopic examinations provided the control biopsies. Patients with EoE and dysphagia had greater esophageal eosinophilia than those with basal cell hyperplasia alone. All normal, EoE, and dysphagia patients had blood drawn into citrate-coated tubes at Tulane University School of Medicine (TUSOM) during a scheduled endoscopy (before and after the treatment of EoE patients). Fresh biopsy samples were collected in RNA buffer for RNA isolation. A centrifuge was used to extract blood plasma and preserve it at −20 °C for VIP analysis. Permission was obtained from patients and TUSOM IRB committee (IRB-approved Study Number: 512127; Study Title: Human Eosinophilic Disorders in Health and Disease Year 2013–2018). Patient and treatment information for normal, EoE, and dysphagia subjects is reported in Appendix A.

### 2.2. Real-Time PCR (RT-PCR) Analysis of RNA Isolated from Esophageal Biopsies of Normal Individuals and EoE Patients

RNA was extracted from esophageal biopsies taken from both healthy controls and EoE patients using a commercially available RNA isolation kit (Qiagen, Hilden, Germany). Following the protocol provided by the manufacturer, reverse transcription was performed on RNA samples (500 ng) from healthy individuals and patients with EoE using iScript reverse transcriptase (Bio-Rad, Hercules, CA, USA). Using IQ5 (Bio-Rad, Hercules, CA, USA) real-time PCR, in which human GAPDH serves as an internal reference, we determined the relative mRNA expression of VIP and VIP receptors in esophageal samples. All primer sequences used for RT-PCR are provided in Appendix A.

### 2.3. Generation of C-kit Neutralized ΔdblGATA Gene-Deficient (ΔdblGATA), IL-13 Transgenic Mice

ΔdblGATA/CC-10-IL-13 transgenic mice were generated by breeding CC-10-IL-13 transgenic mice with ΔdblGATA mice, which are deficient in eosinophils [36]. The resulting F1 generation was subsequently interbred transgenically and ΔdblGATA in F2 offspring was screened. Eosinophil-deficient doxycycline (DOX)-regulated CC-10-IL-13 (ΔdblGATA/CC-10-IL-13) transgenic mice were treated with five doses of 1 mg of anti-c-kit antibody (CD117, clone 2B8), with the first dose administered intravenously and the subsequent doses intraperitoneally. This is based on our pilot study that showed a regimen of 5 doses of 1 mg depletes the esophageal mast cell accumulation in DOX-exposed CC10-IL-13 transgenic mice. The CD117 (2B8) antibody was previously used to deplete bone marrow mast cells in mice [37]. Control ΔdblGATA/CC-10-IL-13 transgenic mice received similar doses of isotype-matched IgG antibody. This approach was a part of a study exploring the impact of eosinophils and mast cell deficiency effects on esophageal muscle contraction and relaxation activity (motility dysfunction) in EoE.

### 2.4. Eosinophil Migration Assay

The chemoattractant behavior of VIP on eosinophils was examined in vitro using 24-well Transwell units with 5 μm porosity polycarbonate filters (Corning Inc., Corning, NY, USA) according to a previously published protocol [38]. Human blood eosinophils were treated with anti-human CCR3 and anti-human Siglec-8 antibodies for 45 min, washed, and sorted with FACS. Purified human eosinophils (10^5^ cells/well) in HBSS, pH 7.2 (Life Technologies, Carlsbad, CA, USA), were placed in the upper chamber, and the recombinant VIP (1, 10, 100, and 500 ng/mL) was added to the lower chamber. Positive controls included eotaxin-2 (200 ng/mL), an eosinophil chemoattractant. The Transwell unit was at 37 °C for 4 h in 95% humidified air with 5% CO_2._ After 4 h, the bottom chamber media was centrifuged at 250 g, and cells were resuspended in PBS and counted using a hemocytometer. An assay was carried out in duplicate wells and repeated three times. The eosinophil migration index is the ratio of VIP-induced eosinophil migration to medium control movement.

### 2.5. Flow Cytometry Analysis of VIP and VIP Receptors on Blood Eosinophils

Human and mouse eosinophils were analyzed using flow cytometry for VIP, VPAC-1, VPAC-2, and CRTH2 receptors. Florescence-tagged antibodies including anti-hCCR3 (FITC, Catalog #310719 Bio Legend, San Diego, CA, USA), anti-hSiglec-8 (PE, Catalog 347103 Bio Legend), anti-hVIP (LS BIO, Catalog LS-B3634-200), anti-hVPAC-1 (Abcam, Catalog ab183312, Waltham, MA, USA) anti-hVPAC-2 (Thermo Fisher, Catalog AVR-002-200UL, Waltham, MA, USA), and anti-hCRTH2 (PE, Catalog 12-2949-42, eBioscience, Waltham, MA, USA) were used to stain blood cells. We used fluorescence-tagged antibodies or primary and secondary antibodies tagged with fluorescent IgG (Santa Cruz Biotechnology, Biolegend, or eBioscience), depending on availability. Labeled IgG antibodies were isotype controls. FlowJo v10 software and a FACS Calibur (BD Biosciences, San Diego, CA, USA) were used for the FACS analysis.

### 2.6. Mouse Bone Marrow Cell-Derived Mast Cell (BMMC) Generation and VIP Receptor Analysis

Flow cytometry was used to assess CRTH2 and VPAC-2 receptor expression on BMMCs. In brief, bone marrow progenitors were cultured in vitro to generate BMMCs in a medium containing 10 ng/mL recombinant murine SCF (PeproTech) and recombinant murine IL-3 (PeproTECH) with 20% fetal bovine serum (FBS), 100 U/mL penicillin, and 100 g/mL streptomycin. For four weeks, 2 × 10^6^ cells were grown in rIL-3 (10 ng/mL)-containing medium with three-day intervals between media and cytokine changes. Mast cell population was detected with flow cytometry using anti-mouse c-Kit (Biolegend) and anti-mouse FCεRI (Biolegend) antibodies on the produced mast cells (2 × 10^5^ cell/tube) [39]. Mast cells obtained from bone marrow were stained using anti-CRTH2 antibody (proSci) and an FITC-labeled anti-rabbit IgG (Biolegend) secondary antibody. The isotype control employed was an IgG antibody that matched the labeled IgG antibody. To identify the VPAC-2 receptor on mast cells, we stained them with anti-mouse c-Kit (Bio Legend, Catalog 105801), anti-mouse FCεRI (eBioscience, Catalog # 17-5898-82), and anti-VPAC-2 receptor (Abcam, Catalog ab246352) antibodies, using the corresponding labeled IgG antibody as an isotype control. The FACSCalibur (BD Biosciences) flow cytometer was used for the analysis, and results were analyzed with FlowJo.

### 2.7. Human Bone Marrow Culture

Human BMCs were purchased from Lonza (Catalog #: 1M-125, Morristown, NJ, USA). The cells were cultured and incubated with IL-3 for 5 days in specialized media (Lonza, Morristown, NJ, USA hematopoietic progenitor growth medium).

### 2.8. Immunofluorescence Staining of Human EoE Esophageal Biopsies for Eosinophils, Mast Cells, NERVE Cells, VIP, and CRTH2 Receptor

Esophageal biopsy sections from healthy individuals and EoE patients were formalin-fixed and paraffin-embedded, and immunofluorescence staining was used to examine CRTH2 receptor expression on eosinophils and VPAC-2 and CRTH2 receptors on mast cells and eosinophil accumulation near nerve cell-derived VIP. In brief, 0.3% hydrogen peroxide in methanol was used to quench endogenous peroxide in the paraffin-embedded sections of esophageal biopsies fixed in formalin, and then, antigen retrieval was performed with pepsin and blocked with 3% goat serum to decrease non-specific binding. Eosinophils were detected with immunostaining using a rat anti-MBP primary antibody (purchased from Mayo Clinic, Phoenix, AZ, USA), followed by FITC-labeled anti-rat IgG as a secondary antibody. Eosinophil accumulation around the nerve cells in esophageal samples was measured using a rabbit-raised anti-PGP9.5 primary antibody (Cell Signaling) and an FITC-labeled anti-rabbit IgG secondary antibody (Biolegend). CRTH2 expression on eosinophils was detected using rabbit-raised anti-CRTH2 (ProSci, Fort Collins, CO, USA) and FITC-labeled anti-rabbit IgG (Biolegend) as a secondary antibody. A goat anti-VIP antibody was used to identify VIP-expressing nerve fibers, while a secondary FITC-anti goat IgG antibody was employed to identify VIP-producing nerve cells. CRTH2 expression on mast cells was also detected in esophageal biopsies using anti-CRTH2 antibody (ProSci) and PE-labeled anti-rabbit IgG (Biolegend) as secondary antibodies, in addition to anti-tryptase antibody and PE-labeled anti-mouse IgG (Bio-Rad) for mast cell detection. To confirm each antibody’s specificity, we used anti-IgG in place of respective antibodies on each tissue section, mounted them on slides using DAPI-mounting material (Thermo Fisher Scientific), and photographed single- or double-immunostained tissue sections using an Olympus BX51 microscope. The quantification of eosinophils: Eosinophils were quantified by counting the anti-MBP, tryptase, or any receptor-positive cells in the epithelial mucosa, lamina propria, and muscularis mucosa of the esophageal tissue section using digital morphometric analysis (Luminera Corporation, Infinit Analyze 610, Ottawa, ON, Canada). Tissue samples were taken from the same area in each set of mice, and 4–6 random sections/mouse were analyzed for each stained tissue. Results were expressed as the respective positive cells or receptor/mm^2^ human EoE biopsies from distal and proximal inflamed areas were processed, and tissue slides were similarly stained and counted morphometrically. 

### 2.9. CRTH2 Antagonist Treatment in Experimental EoE

The Jackson Laboratory (Bar Harbor, ME, USA) provided the specific pathogen-free BALB/c mice. Age- and sex-matched mice (6–8 weeks) were used in all trials. EoE was induced using similar methods as in the earlier experiments [40,41]. Briefly, mice were anesthetized with a light dose of isoflurane (Iso-Flo; Abbott Laboratories, North Chicago, IL, USA), then administered 100 μg *Aspergillus fumigatus* (Greer Laboratories, Lenoir, NC, USA) in 50 μL normal saline (or 50 μL normal saline alone for control mice) intranasally three times weekly for three weeks while in a supine position. Animals were administered an injectable dose of CRTH2 antagonist (OC000459) (Cayman Chemical, Ann Arbor, MI, USA) every other day leading up to the final *Aspergillus* challenge. The mice were sacrificed 24 h after the final intranasal allergen or saline exposure. Anti-MBP immunostaining was used to examine eosinophils in esophageal tissue slices, and chloroacetate esterase labeling was used to examine mast cells, both in accordance with the previously reported techniques [31,42]. Procedures involving animals were approved by the Tulane Institutional Animal Care and Use Committee and complied with the NIH regulations.

### 2.10. Analysis of the Contractility and Relaxation of the Esophageal Circular and Longitudinal Muscle Layers in Experimental EoE

We tested esophageal circular and longitudinal muscle contractility in CC10/IL-13 mice and wild-type mice using a described previously technique [38]. In brief, the removed esophagus was tested for longitudinal muscle contractility. The esophagus was kept in a physiological salt solution after dissection. PSS (mmolar/l) had 118 NaCl, 4.73 KCl, 1.2 MgSO_4_, 0.026 EDTA, 1.2 NaH_2_PO_4_, 2.5 CaCl_2_, 5.5 glucose, 25 NaHCO_3_, and a pH of 7.4 when bubbled with 95% O_2_/5% CO_2_ at 37 °C. To measure longitudinal isometric force, a Differential Capacitor Force Transducer (Harvard Apparatus, Holliston, MA, USA) was mounted to the esophageal ring and longitudinal section. Adjusting the esophagus length to the maximum isometric contractions produced the optimal tension for force output in experiments. To ensure reproducible contractions, basal measurements were taken after 1 h of equilibration after two stimulation and relaxation cycles with 15 mM KCl. Concentration–isometric force relations were generated by adding carbamylcholine (1 nM to 10 μM) at 5 min intervals. Concentration–relaxation relations were generated at 80% of the highest isometric force from the concentration–force relation to avoid agonist saturation and compare all esophagi at the same activation. Isoproterenol (1 nM to 10 μM) was added at 5 min intervals to evaluate relaxation. A model MP100 Data Acquisition System (Biopac Systems, Goleta, CA, USA) was utilized at 100 Hz. Data analysis was performed with Biopac AcqKnowledge (Biopac Systems). Force was normalized by dividing by the cross-sectional area, approximated as wet weight/length for longitudinal preparations or 2* (wet weight/circumference) for rings. Concentration–response relations were fitted with a logistic function using the Origin (v9.9) software, and differences in parameters between transgenic and wild-type mice were analyzed with *t*-test. Data are expressed as means ± standard deviation (SD).

We evaluated the relaxation of esophageal longitudinal muscles following doxycycline (DOX)-regulated CC10/IL-13-induced esophageal inflammation following an experimental protocol similar to the contractility experiment. The basal measurement was sampled for 100 s following the exposure of the esophageal segments to two cycles of stimulation with 15 mM KCl. The peak contraction force response was generated in response to CCh (1 μM), and the subsequent concentration-dependent relaxation of muscles was generated using the cumulative addition of isoproterenol (1 nM to 1 μM) to determine the relaxation force of the circular and longitudinal muscle layers of control and inflamed esophagi. The data were analyzed using the companion Biopac Acqknowledge v3.9 software.

### 2.11. Statistical Analysis

All data are presented as the mean ± standard deviation (SD). Statistical significance was determined using the SAS software (version 9.4 or higher) in a Windows environment (SAS Software, Cary, NC, USA). Data involving two groups was analyzed using a Mann–Whitney test with the Bonferroni correction, and for more than two groups, Kruskal–Wallis test was used and considered statistically significant (*p* < 0.05).

## 3. Results

### 3.1. Analysis of VIP and VIP Receptors (VACP1, VAPC2, and CRTH2) on Eosinophils and Mast Cells and the VIP Receptor’s Role in the Migration of Both Inflammatory Cells 

The role of nerve cell-derived VIP has been implicated in promoting eosinophilic inflammation in allergic diseases [43,44,45,46], so we hypothesized that VIP, along with eotaxin, may have a role in the accumulation of eosinophils and mast cells in each segment of the esophagus in human EoE. We first examined VIP and its receptors in human blood eosinophils using anti-CCR3 and anti-Siglec8 antibodies followed by anti-VIP, anti-VAPC1, anti-VACP2, and anti-CRTH2 receptors (Figure 1a–d). Since mast cells are not present in blood, we generated in vitro mast cells in response to 10 ng/mL of rIL-3 (PeproTECH)-treated human bone marrow cells (Lonza, USA) per the method described earlier [47]. Mast cells were examined using anti-ckit, anti-FcεRI^+^, and VIP and its receptors on mast cells using anti-VIP, anti-VAPC1, anti-VACP2, and anti-CRTH2 antibodies. Flow cytometry analysis detected the expression of VAPC2 and CRTH2 receptors on anti-cKit^+^/anti-FcεRI^+^ double-positive mast cells, but no expression of VIP or VACP1 receptor (Figure 1e–h) was observed. A representative dot plot flow cytometry analysis of anti-CCR3^+^ and anti-Siglec8^+^ eosinophils (Appendix A) and anti-cKit^+^ and anti-FcεRI^+^-positive mast cells (Appendix A) are included below. We kinetically examined human eosinophil motility in response to different concentrations of VIP recombinant protein (0, 1, 10, 100, 500 ng/mL) and another eosinophil chemoattracting recombinant protein, PGD2 (500 ng/mL), using Transwell chemotactic chambers (Creative Bioarray). The recombinant protein of established eosinophil chemokine eotaxin-3 (200 ng/mL) was used as a positive control. Eosinophil chemoattraction in response to VIP showed a dose-dependent increase in eosinophil motility, like eotaxin-3 and PGD2 (Figure 1i). We determined the in-vitro specificity of VIP receptor CRTH2 in eosinophil motility using anti-CRTH2 antibody. We found a significantly reduced in vitro eosinophil migration index of anti-CRTH2-pretreated eosinophils compared to non-treated eosinophils in response to 200 ng/mL VIP. Anti-CRTH2-treated eosinophils did not restrict their migration index in response to 200 ng/mL eotaxin-3, although a similar motility index was observed (Figure 1j(i)). Our analysis indicated that mast cells also migrated in response to VIP, and anti-CRTH2 pretreatment restricted mast cell migration in response to 200 ng/mL VIP (Figure 1j(ii)). The migration assay was performed in three separate experiments, and the average is presented as mean ± SD.

### 3.2. Analysis of Nerve Cell-Derived Mediator VIP Protein in the Blood and mRNA Levels of Esophageal Biopsies along with the Expression of VIP Receptors VAPC1, VACP2, and CRTH2 in Normal, EoE, and Dysphagia Patients

The gastrointestinal tracts of human and rodents, including the esophagus, exhibit the VIP-mediated relaxation of the lower esophageal sphincter [48]. We examined the induced blood VIP protein levels and esophageal biopsy mRNA levels of VIP and VIP receptors (VACP1 and CRTH) in dysphagia patients compared to normal individuals (normal vs. EoE, *p* < 0.001, *n* = 14–18; Figure 2b–d). We also statistically examined the correlation of VIP with CRTH2 receptor mRNA, *p* < 0.0001 (Figure 2e), and eosinophil count relative to CRTH2 receptor mRNA levels in the biopsies of EoE patients, (*p* < 0.005, *n* = 17–18; Figure 1f). r^2^ values were calculated using the Mann–Whitney test with the Bonferroni correction. Data on blood protein (*n* = 7–13) and mRNA (*n* = 13–18) levels are expressed as mean ± SD. 

### 3.3. Detection of CRTH2 on Eosinophils and VAPC2 Receptor on Mast Cells in Human EoE Biopsies

We examined CRTH2 and VAPC2 receptor expression on tissue-accumulated human eosinophils and mast cells by performing anti-MBP and anti-CRTH2 for eosinophils and anti-tryptase and anti-VAPC2 for mast cells on the biopsy sections of human EoE patients (*n* = 7–8; Figure 3a–f). Tissue accumulated both anti-MBP positive eosinophils and anti-tryptase-positive mast cells, colocalized with anti-CRTH2 and anti-VAPC2 receptor, respectively. The representative photomicrographs were captured using computer-assisted camara and software (Figure 3c,f). These histologically analyzed data validated our in vitro CRTH2 and VAPC2 expression data on eosinophils and mast cells. 

### 3.4. Analysis of Eosinophils and Mast Cell Accumulation in All Segments of Human Esophageal Biopsies in Human EoE

We set out to determine if VIP is responsible for the accumulation of eosinophils and mast cells in all sections of the epithelial mucosa, including the muscularis mucosa, which regulates the motility function in the esophagus. An accumulation of eosinophils and mast cells is implicated in the onset of motility dysfunction in experimental and human EoE [31,49,50]. Next, we performed deep biopsies using specialized endoscopy forceps to obtain all the segments of human esophageal biopsy, including epithelial, subepithelial muscular, and muscularis mucosae of EoE patients (Appendix A). We performed H&E staining on these deep biopsies and found eosinophil accumulation in all segments, including both epithelial and muscularis mucosae (Appendix A). 

### 3.5. Esophageal Eosinophils and Mast Cells Are Detected in Nearby Nerve Cells Derived from VIP and Expressed in the Epithelial Mucosa in Human EoE

An earlier study implicated a certain family of proteins (PGP 9.5) that is confined to neural and neuroendocrine cells [51]. Since VIP is the product of endocrine cells [52,53,54], we tested the hypothesis that VIP is expressed by nerve cells in the esophagus of human EoE, and that nerve cells derived from VIP may have an important role in eosinophil accumulation in and beyond the epithelial mucosa in human EoE, apart from eotaxin-3. Previous reports indicate that the epithelial cell-derived eotaxin-3 is highly induced in EoE and has chemoattractant activity for eosinophils [55]; however, we found that in vitro motility of eosinophils and mast cell motility in response to VIP are similar to those of eotaxin-3. We performed the immunofluorescence staining of anti-VIP and anti-PGP 9.5 for nerve cells derived from VIP, anti-MBP for eosinophils, and anti-tryptase for mast cells in human deep biopsy tissue sections containing all esophageal segments. Epithelial mucosa underwent immunofluorescence staining with anti-MBP and anti-VIP to detect eosinophils nearby VIP-expressing cells, anti-tryptase and anti-VIP to detect mast cells nearby VIP-expressing cells, and anti-PGP 9.5 to detect VIP being expressed by nerve cells in human EoE biopsies (Figure 4). A merged photomicrograph showing anti-MBP and anti-VIP stained with DAPI-mounted tissue sections of epithelial mucosa showed the accumulation of eosinophils nearby nerve cells (Figure 4a–d); anti-VIP expressed in anti-PGP9.6-expressing nerve cells in DAPI-mounted tissue sections identified VIP-expressing nerve cells (Figure 4e–g), and anti-tryptase and anti-VIP staining with DAPI showed the accumulated mast cells nearby VIP-expressing nerve cells in the epithelial mucosa of human esophageal EoE biopsies (Figure 4h–j). Some eosinophils were also detected in the EoE biopsies where no VIP expression was detected (Figure 4d), suggesting the possibility that other chemoattractants like eostaxin-3 may also contribute to eosinophil accumulation in the epithelial mucosa. A morphometric analysis was performed for eosinophils near (distance < 1 μm) and far (distance > 1 μm) from nerve cells, and all mast cells were colocalized with VIP-expressed cells in esophagus epithelial region in the biopsies of human EoE (Figure 4k). The representative photomicrographs are presented as 40× (a) and 400× (b–j). Data are expressed as mean ± SD, *n* = 6–7. 

### 3.6. Eosinophils and Mast Cells Accumulate in the Muscularis Mucosa of Human EoE Patients

We also found eosinophil and mast cell accumulation in the muscularis mucosa region following the immunofluorescence staining of anti-MBP, anti-tryptase, and anti-VIP in deep biopsies of EoE patients. A merged photomicrograph of anti-MBP and anti-VIP stained with DAPI-mounted tissue sections shows eosinophils nearby VIP-expressing nerve cells (Figure 5a–d); similarly, anti-tryptase and anti-VIP staining with DAPI-mounted tissue sections showed mast cell accumulation adjacent to VIP-expressing nerve cells (Figure 5e–h). We also stained the tissue sections with anti-PGP 9.5/anti-VIP and anti-tryptase/anti-VAPC2, which allowed us to observe the accumulation of VIP positive cells and anti-tryptase and anti-VAPC mast cells nearby nerve cells (Figure 5h–m). A morphometric study of eosinophils near and far from nerve cells revealed the colocalization of mast cells with VIP-expressed cells in the esophageal epithelial region of human EoE samples (Figure 5n).

### 3.7. Doxycycline (DOX)-Regulated CC10-IL-13-Overexpressed Animal Model of Experimental EoE Has Similar Eosinophils and Mast Cells in Each Esophagus Segment near VIP-Expressing Nerve Cells

The activation and degranulation of eosinophils and mast cells have been implicated in the induction of esophageal functional abnormalities, including stricture and motility dysfunction, in human EoE [56]. Since the mechanistic and preclinical studies of esophageal function cannot be conducted on human EoE patients, we used a doxycycline (DOX)-exposed CC-10 IL-13 transgenic chronic EoE mouse model [57,58] generated by crossing two different transgenic mice using the construct presented in Appendix A. We found that a similar VIP-associated mechanism is also operational for eosinophil and mast cell accumulation in and beyond the epithelial mucosa of an experimental DOX-regulated IL-13-overexpressed chronic mouse model of EoE. We showed that eosinophils and mast cells accumulated nearby VIP-expressed nerve cells (distance < 1 μm) from three weeks of DOX-exposed CC-10-IL-13 in the muscularis mucosa of a mouse model of experimental EoE. Eosinophils and mast cells also accumulated in the immunofluorescence staining of anti-MBP, anti-tryptase, and anti-VIP in muscularis mucosa in the tissue sections of DOX-exposed CC-10-IL-13 mice (Figure 6a(i–iv)). Previously, we showed that CC10-IL-13 mice accumulate eosinophils and mast cells in each segment of the esophagus [56]; here, we present evidence that these mice also accumulate eosinophils and mast cells near nerve cell-expressed VIP in the muscularis mucosa. A merged photomicrograph of DAPI-mounted anti-MBP and anti-VIP-stained tissue sections shows eosinophils nearby VIP-expressing nerve cells (Figure 6a(iv)) and anti-tryptase and anti-VIP mast cells nearby nerve cells in the esophageal tissue sections of our DOX-exposed CC-10-IL-13 mouse model of EoE (Figure 6b(i–iv)). Next, we stained these tissue sections with anti-PGP 9.5/anti-VAPC2 and anti-PGP 9.5/anti-CRTH2 and found the accumulation of VIP receptor-positive cells nearby nerve cells (Figure 6c(i–iv),d(i–iv)). Doxycycline’s short-term use has no effect on mice; its function is to induce a particular gene inserted to the promoter. WT mice exposed to DOX do not show any tissue pathological abnormalities, but changes in gut microbiome and insulin levels were observed after long exposure in transgenic mice [59,60]. These data confirm that this mouse model mimics human EoE and is suitable for investigating the mechanism underlying eosinophil and mast cell accumulation in tissue beyond the epithelial mucosa that promotes functional abnormalities, like motility disorders, in human EoE. Photomicrographs are presented at 400× original magnification.

### 3.8. Longitudinal Esophageal Muscle Contraction and Relaxation Activity in Mast Cell-Neutralized Eosinophil-Deficient DOX-Regulated IL-13-Overexpressed Mice Compared to DOX-Regulated CC10-IL-13-Overexpressed Mouse Model of EoE

We have shown that eosinophil-deficient ΔdblGATA-CD2-IL-5-overexpressed mice do not show improvement in esophageal longitudinal muscle contraction dysfunction [57]. Next, we examined whether both eosinophils and mast cell deficiency in a DOX-regulated CC10-IL-13 chronic mouse model of EoE improve the impaired esophageal longitudinal muscle contraction and relaxation compared to doxycycline (DOX)-exposed CC-10-IL-13 mice. It has been shown that CC10-IL-13-overexpressed mouse esophagus accumulates inflammatory cells associated with fibrotic responses in the esophagus [61] and that esophageal muscle contractility with chronic esophageal inflammation affects esophageal motility [57]. We generated DOX-regulated eosinophil-deficient ΔdblGATA-CC10-IL-13 transgenic mice by breeding ΔdblGATA mice with DOX-regulated CC10-IL-13 mice as described previously [58], and mast cells were neutralized by intraperitoneally injecting anti-C-kit antibody (CD117, clone 2B8, BioLegend, San Diego, CA, USA) 1 mg/mouse/week as per the protocol shown in Appendix A. We examined the carbachol-induced esophageal muscle contractility in CC-10-IL-13 mice exposed to DOX for 3 weeks, DOX-exposed C-kit-neutralized ΔdblGATA/CC-10-IL-13 mice with esophagi ex vivo, and in no-DOX-exposed CC-10-IL-13 mice. We observed that DOX-exposed CC10-IL-13 mice developed more force in esophageal muscle contraction in response to the cumulative additions of carbachol compared to C-kit-neutralized ΔdblGATA/CC-10-IL-13 (both eosinophil- and mast cell-deficient) mouse models of EoE, the latter of which showed significantly reduced force in contraction of esophageal muscles. No-DOX-exposed CC-10-IL-13 mice showed highly reduced force in response to carbachol (Figure 7a). We also examined 3-week-exposed DOX-regulated CC10-IL-13-overexpressed mice and C-kit-neutralized DOX-regulated ΔdblGATA/CC-10-IL-13 mice, with longitudinal esophagus muscles precontracted with carbachol to achieve the maximal response (6.5 ± 0.13 MN/mm^2^), then isoproterenol was cumulatively added as shown in Figure 7b. We observed that the relaxation of isometric force on the esophageal smooth muscle of DOX-exposed C-kit-neutralized ΔdblGATA/CC10-IL-13-overexpressed mice was significantly reduced compared to CC10-IL-13-overexpressed mice. Both DOX-regulated CC10-IL-13-overexpressed mice and C-kit-neutralized ΔdblGATA/CC10-IL-13-overexpressed mice showed esophageal relaxation in response to isoproterenol in a concentration-dependent manner (Figure 7b). Staining tissue sections with chloroacetate esterase revealed that anti-c-Kit-neutralized eosinophil-deficient IL-13-overexpressed (ΔdblGATA/CC10-IL-13) mice are also deficient in mast cells compared to WT or CC10 IL-13-overexpressed mice (Figure 7c(i–iv)). This suggests that the accumulation of both eosinophils and mast cells is responsible for motility dysfunction in EoE. Data are expressed as mean ± SEM; n = 3 experiments.

### 3.9. CRTH2 Antagonist Treatment Improves Impaired Longitudinal Esophageal Dysfunction in a DOX-Regulated CC10-IL-13-Overexpressed Mouse Model of EoE

We have shown that both eosinophils and mast cells express VIP receptor CRTH2, and in vitro CRTH2 antagonists restricted the migration of both eosinophils and mast cells. Thus, we decided to determine whether the pharmacological CRTH2 receptor antagonist treatment of eosinophils improves esophageal longitudinal muscle contraction and relaxation dysfunction in DOX-inducible CC10-IL-13 mice. We examined the effect of ex vivo CRTH2 antagonist treatment on motility in esophagi of DOX-inducible CC10-IL-13 mice in response to the cumulative additions of carbachol (CCh) and isoproterenol (ISO), respectively. Our analysis indicated that the esophagi of CC10-IL-13 mice exposed to DOX for 3 weeks developed more force in esophageal muscle contractility in response to cumulatively increased carbachol concentrations, compared with CRTH2 antagonist-treated DOX-exposed CC10-IL-13 mice (Figure 8a). This carbachol-induced contractility suggests a concentration-dependent force that is reduced following CRTH2 antagonist treatment. The maximum contractility isometric force of 7.23 ± 0.15 mN/mm^2^ for this group was significantly greater than that of the CRTH2 antagonist-treated CC10-IL-13 mice, (4.71 ± 0.24 mN/mm^2^). No-DOX CC10-IL-13-overexpressed mice showed highly reduced force in response to contractility, suggesting a concentration-dependent force that is reduced following CRTH2 antagonist treatment. The maximum contractility isometric force of 7.23 ± 0.15 mN/mm^2^ was significantly greater than that of the CRTH2 antagonist-treated CC10-IL-13 mice, (4.71 ± 0.24 mN/mm^2^). No-DOX CC10-IL-13-overexpressed mice showed highly reduced force in response to carbachol (Figure 8a). The longitudinal preparations of esophageal muscles were precontracted with carbachol to achieve 6.3 ± 0.39 mN/mm^2^ of the maximal response. Following the cumulative additions of isoproterenol, the relaxation of isometric force in esophageal smooth muscle from CRTH2 antagonist-treated DOX-exposed CC10-IL-13-overexpressed mice showed significantly reduced muscle relaxation compared to CC10-IL-13-overexpressed mice. Both no-DOX-regulated CC10-IL-13-overexpressed mice and CRTH2 antagonist-treated CC10-IL-13-overexpressed mice showed significantly reduced esophageal longitudinal muscle isometric tension in response to isoproterenol (Figure 8b). This indicates that CRTH2 antagonist treatment may be a promising treatment strategy to improve motility dysfunction in EoE. The immunofluorescence analysis of eosinophils expressing CRTH2 and mast cells expressing CRTH2 were examined, and the expression is shown in the esophageal tissue sections of CC10-IL-13-overexpressed mice (Figure 8c(i–iv),d(i–iv)).

## 4. Discussion

Eosinophilic esophagitis (EoE) is a clinicopathological disease associated with symptoms similar to gastroesophageal reflux disease (GERD). It is associated with the excessive accumulation of eosinophils and mast cells in the esophageal epithelial and muscular mucosae, including muscularis mucosa [4,62,63]. The esophageal accumulation of eosinophils and mast cells, along with degranulation in the esophagus, are characteristic features that promote chronic pathogenesis, including dysphagia and esophageal motility dysfunction, in both EoE and GERD [9,64,65,66]. Clinical studies have suggested that the prevalence of these disorders, especially EoE, is increasing globally [67,68]. Several cytokines and chemokines are implicated in eosinophil- and mast cell-mediated pathogenesis in experimental and human EoE [6,31,69]. The chemokine eotaxin-3 is also significantly correlated with esophageal eosinophilia in human EoE biopsies [55,70]. Despite this established correlation, no direct evidence of the role of eotaxin-3 in human EoE has been established [71]. It is possible that other chemotactic factors may also be involved in the activation of eosinophils, mast cells, and the esophageal mucosa of human EoE. The focus of this study is to establish the molecular events that occur in association with eotaxin-3 in EoE and its relationship to the recruitment, accumulation, and degranulation of eosinophils and mast cells characteristic of EoE. We present evidence for a novel neuroimmune pathway involved in the recruitment of eosinophils and mast cells in the epithelial mucosa and beyond into the muscular mucosa in EoE [72]. We show that eosinophils indeed accumulate in the muscular mucosa of the esophagus and other gastrointestinal segments nearby nerve cells [56]. These findings indicate that the interaction of nerve cell-derived VIP and VIP receptors present on eosinophils and mast cells is critical in the induction of eosinophilic inflammatory gastrointestinal diseases. These observations are in accordance with ultrasonographic evidence that eosinophils accumulate in the submucosa and muscular mucosa [49,50]. 

We provide evidence that the nerve cell-derived mediator VIP attracts mast cells and eosinophils to accumulate in the muscular mucosa and may promote esophageal functional abnormalities like motility dysfunction in an experimental model of EoE. Eosinophils accumulate in nearby nerve cells, and the nerve cell-derived polypeptide VIP is implicated in the recruitment, chemotaxis, and degranulation of eosinophils and mast cells, suggesting that VIP mediates chemotaxis of these immune cells in the esophagus [28,73,74,75]. These findings validate reports on VIP response to eosinophil accumulation in the lung [29,76], which is also observed in other eosinophil-associated allergic diseases [43,77,78]. Our data show that VIP is highly induced in EoE patients with dysphagia, and both eosinophils and mast cells express the CRTH2 receptor in vitro and in biopsies of EoE patients. We also present evidence that CRTH2-expressed eosinophils accumulate adjacent to VIP-producing nerve cells in human EoE. The migration of eosinophils in vitro against VIP provides direct evidence that VIP, like eotaxin-3, is indeed involved in eosinophil recruitment. Mechanistically, we show that anti-CRTH2 pretreatment restricts eosinophil and mast cell motility in both in vitro and in vivo experimental models of EoE. We also show that VIP-induced eosinophil motility in vitro is specific to the CRTH2 receptor, since anti-CRTH2 pretreatment did not affect eosinophil motility in response to eotaxin-3. We also found several eosinophils present in the esophageal mucosa away from nerve cell-derived VIP, indicating that both VIP and eotaxin-3 are important for eosinophil accumulation in EoE, and motility dysfunction occurs in response to the VIP-induced accumulation of eosinophils and mast cells. We provide direct evidence of VIP as a chemotactic factor for the accumulation of eosinophils and mast cells in EoE. Our findings show that IL-3-derived mast cells in vitro express the VIP receptor CRTH2, which further validates that tissue mast cells express CRTH2 and supports the earlier reports that indicate CRTH2 is expressed on human mast cells [79]. VIP is secreted by neurons in the esophagus where it relaxes the lower sphincter muscles, so we believe there is compelling evidence in favor of human clinical trials of anti-VIP or VIP receptor antagonist therapy to both reduce esophageal eosinophilia and improve esophageal motility dysfunction in EoE and even in GERD. Several other molecules are shown to be induced by VIP in allergic and autoimmune diseases, including TSLP, TGF-β, and VCAM-1, all of which are implicated in the pathogenesis of EoE [80,81,82]. Several studies have shown that targeting eosinophil mast cell receptors or VIP receptors improves the pathogenesis of allergic diseases [32,83,84,85,86,87,88]. 

In this report, we demonstrate that VIP and VIP receptors are induced in human EoE, VIP has chemoattractant activity for eosinophils similar to that of eotaxin 3, and the signaling of VIP in humans is linked to the CRTH2 receptor on eosinophils and CRTH2 and the VPAC-2 receptor on mast cells. We provide evidence that eosinophils accumulate adjacent to VIP-producing nerve cells. The current study identifies a novel and important chemoattractant role of VIP in the accumulation of eosinophils and mast cells that promotes EoE pathogenesis. VIP may be a novel target molecule for chronic EoE therapy, possibly attenuating dysphagia, stricture, and motility dysfunction. Our study suggests the need for a multicenter anti-CRTH2 antagonist or anti-VIP clinical trial to improve quality of life for EoE patients.

## 5. Conclusions

In conclusion, we presented evidence on the significance of the interaction of VIP and VIP receptors CRTH2 and VAPC2 present on eosinophils and mast cells, respectively, in human EoE. Further, we showed that VIP has chemoattractant activity for eosinophils, comparable to eotaxins-3. Additionally, we showed that nerve cells are the source of VIP that **chemoattract** eosinophils in and beyond the epithelial mucosa, including muscular mucosa that controls esophageal motility, and demonstrated that eosinophil and mast cell deficiency prevents the development of motility dysfunction. Lastly, we showed that the CRTH2 antagonist treatment of chronic experimental EoE. Thus, CRTH2 antagonist may be a promising candidate for the treatment of esophageal motility dysfunction in human EoE. Accordingly, we propose a requirement for a multicenter clinical trial involving anti-CRTH2 antagonist or anti-VIP treatment to improve the quality life of EoE.

## Figures and Tables

**Figure 1 cells-13-00295-f001:**
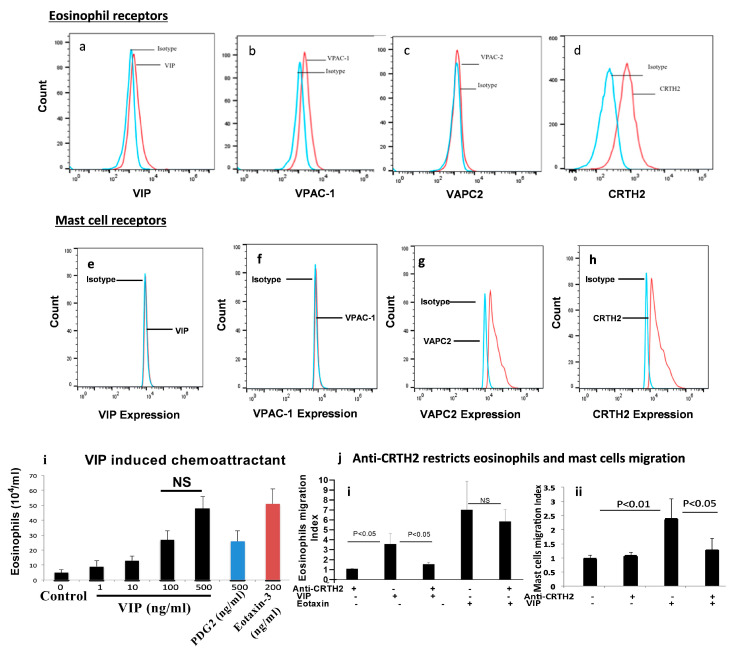
**Analysis of VIP and VIP receptors VPAC-1, VPAC-2, and CRTH2 receptors on human blood eosinophils and in vitro-generated mast cells and their migration in response to VIP**. Anti-VIP, anti-VAPC1, anti-VACP2, and anti-CRTH2 antibodies were used to assess VIP and VIP-associated receptor (VAPC1, VACP2, and CRTH2) expression in human eosinophils with flow cytometry analysis (**a**–**d**). Further, the VIP and VIP receptors (VAPC1, VACP2, and CRTH2) were also analyzed in in vitro-generated mast cells from human bone marrow using flow cytometry (**e**–**h**). Furthermore, the analysis of the chemoattractant assay in response to dose-dependent VIP, PDG-2, and eotaxins-3 (**i**). Anti-CRTH2 antibody inhibited VIP-induced eosinophil migration in vitro but had no effect on eotaxin-2-induced eosinophil migration. These results demonstrate that the interaction between CRTH2 and VIP is essential for eosinophil motility (**j**(**i**)). Anti-CRTH2 antibody inhibits VIP–induced mast cell migration in vitro (**j**(**ii**)). Data are expressed as mean ± SD, *n* = 3 experiments. Migration data are presented as an average of three separate experiments. Data involving two groups were analyzed with a Mann–Whitney test; for more than two groups, Kruskal–Wallis test was used. Significance was set at *p* < 0.05 and *p* < 0.01, NS—no significant respectively.

**Figure 2 cells-13-00295-f002:**
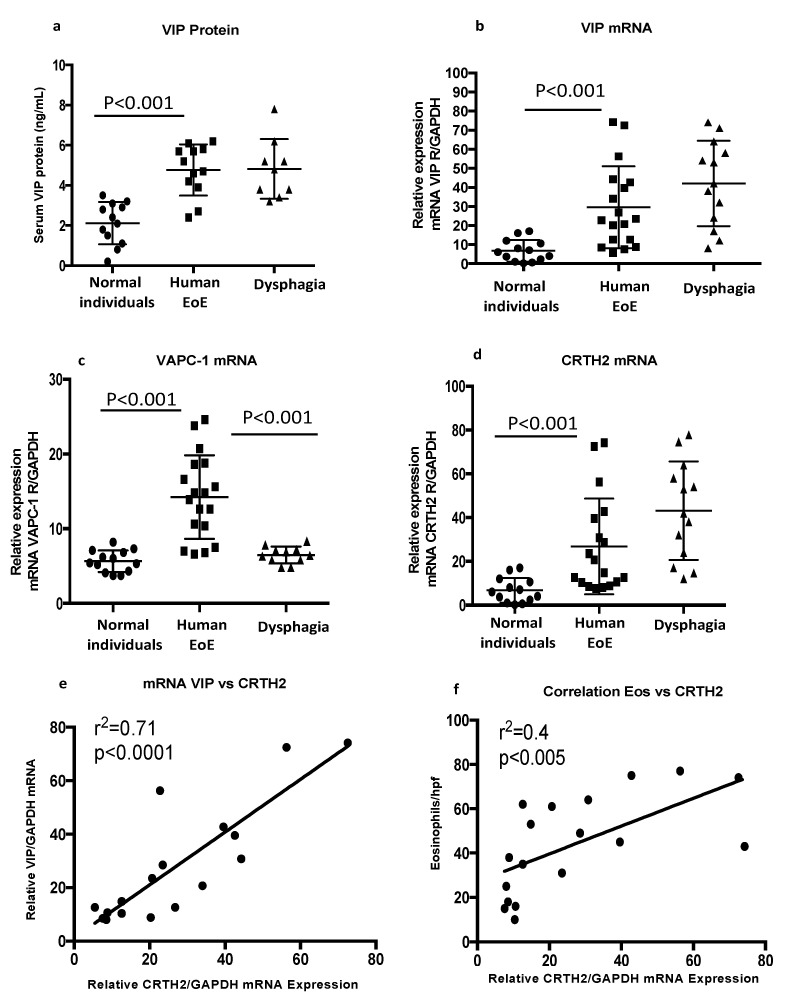
**Transcript analysis of VIP and correlation of VIP-associated receptors in esophageal biopsies of human EoE**. ELISA analysis indicated that expression levels of VIP protein and mRNA (**a**,**b**) and VIP receptors VAPC1 and CRTH2 were higher in EoE and dysphagia biopsies compared to controls (**c**,**d**). A significant statistical transcript correlation between VIP and CRTH2 (*p* < 0.0001, r = 0.71) (**e**), as well as between CRTH2 and eosinophils (*p* < 0.005, r = 0.4), was observed in EoE patients (**f**). Significance was calculated using Mann–Whitney test for two groups, and Kruskal–Wallis for more than two groups, and correlation (r^2^) values were calculated using the Mann–Whitney test with the Bonferroni correction. Data are expressed as mean ± SD, *n* = 8–12. Each data point represents one individual patient.

**Figure 3 cells-13-00295-f003:**
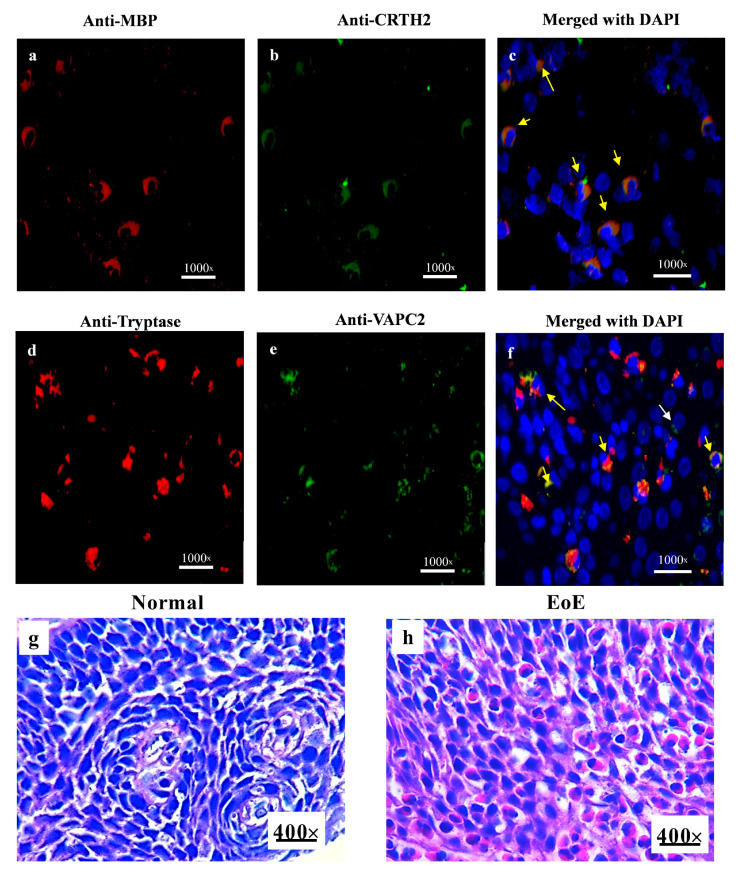
**Human esophageal EoE eosinophils and tissue-accumulated mast cells express CRTH2 and VAPC2 receptors**. Double immunofluorescence staining of esophageal biopsies of EoE patients showed anti-MBP stained eosinophils (**a**) and tryptase for mast cells (**d**); anti-CRTH2, anti-VAPC2-stained CRTH2 and VIP receptor (**b**,**e**). DAPI-mounted photomicrographs show tissue eosinophils expressing CRTH2 receptor and mast cells expressing VAPC2 receptor in human EoE, indicated by arrows (**c**,**f**), H&E-stained EoE patient biopsies show a highly induced eosinophil accumulation in human biopsies compared to very few in normal biopsy tissue sections (**g**,**h**). A total of 7–8 patient biopsies were examined, and 1000× and 400× field/biopsies were analyzed for 3–4 patients. The yellow arrow indicates the positive cells. A randomly selected representative photomicrograph is presented at 400× original magnification.

**Figure 4 cells-13-00295-f004:**
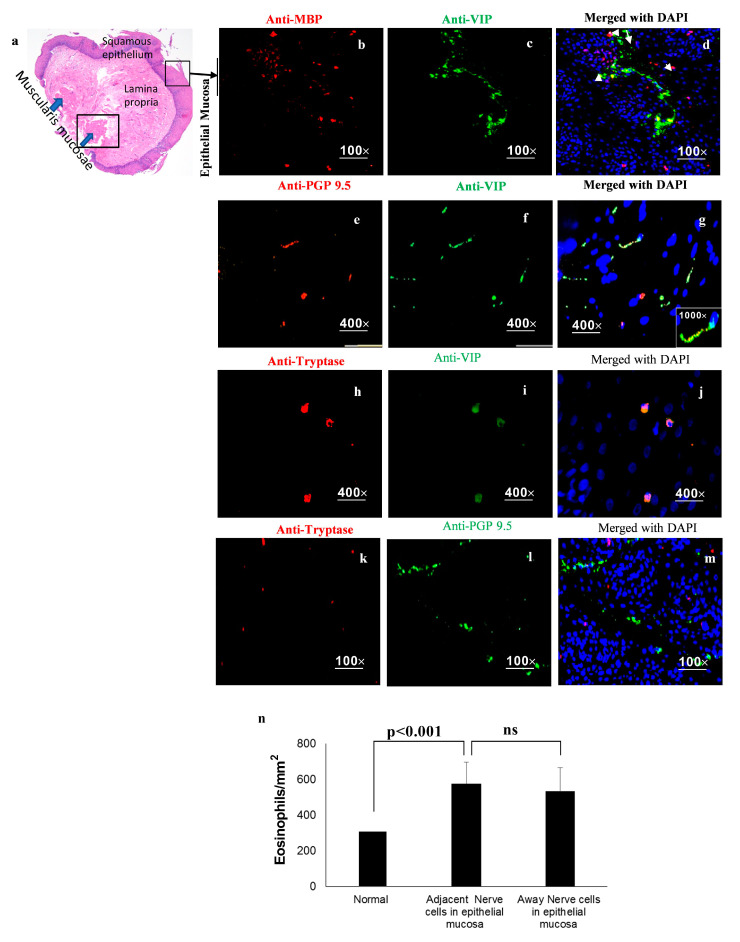
**Immunofluorescence detection of eosinophils nearby nerve cells of epithelial mucosa in EoE patients.** H&E staining of eosinophils in human EoE patients shows all segments including muscularis mucosa of deep esophageal biopsies (**a**). Anti-MBP-stained eosinophils and VIP nerve cells in the esophageal biopsies (stained red and green, respectively; (**b**–**d**)). Anti-PGP9.5-stained nerve cells and VIP are shown with low and high magnification in the esophageal biopsies of EoE patients (green and red, respectively; (**e**–**g**)). The DAPI-merged photomicrograph clearly shows the striking accumulation of anti-MBP-stained eosinophils nearby PGP 9.5-stained nerve cells in esophageal biopsies of EoE patients (**d**,**g**). Arrows indicate the recruitment of eosinophils nearby or on the nerve cells at low (100×; (**d**)) and high (400×; (**g**)) magnification of original photomicrographs of EoE biopsies. Double immunofluorescence of anti-tryptase and VIP was found in the epithelial mucosa of human EOE patient samples (**h**–**j**). Immunofluorescence of anti-tryptase and anti-PGP 9.5 was found in the epithelial mucosa of human EOE patient samples (**k**–**m**). Quantitation of eosinophils indicates that eosinophils accumulated nearby (distance < 1 μm) and away from (distance > 1 μm) nerve cells in the esophageal epithelial mucosa in EoE patients; no eosinophils were detected in control biopsies (**n**). ns—no significant. Data are expressed as mean ± SD, *n* = 6–7 patients, 3–4 hpf field, randomly selected biopsy sections.

**Figure 5 cells-13-00295-f005:**
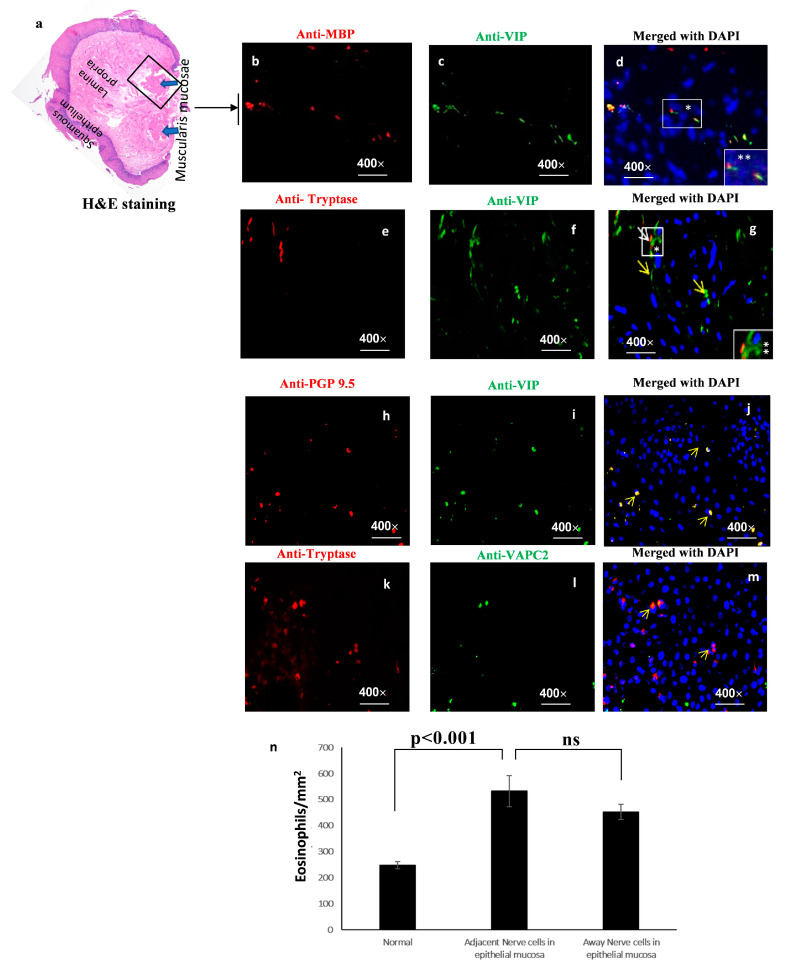
**Immunofluorescence detection of eosinophils nearby nerve cells of muscularis mucosa in human EoE**. Samples from EoE patients were subjected to H&E staining to visualize eosinophils in deep esophageal biopsies (**a**). Esophageal samples were subjected to immunofluorescence labeling using an anti-MBP antibody. Eosinophils (MBP) were stained red and VIP cells green (**b**–**d**). The muscularis mucosa layer from one EoE patient was subjected to double immunofluorescence staining using antibodies against tryptase and vasoactive intestinal peptide (VIP). The resulting images (**e**–**g**) demonstrated the colocalization of these two markers. No eosinophils were observed in biopsies obtained from individuals without EoE. Immunofluorescence staining of anti-PGP 9.5 and anti-VIP, as well as anti-tryptase and the VIP receptor VAPC2, showed a significant difference between EoE biopsies compared to normal biopsies (**h**–**m**). The relative abundance of eosinophils present near nerve cells (distance < 1 μm) versus those located further away (distance > 1 μm) in the esophageal epithelial mucosa of individuals diagnosed with EoE is shown (**n**). Data are presented as mean ± SD of patients (*n* = 6, 3–4 hpf field/biopsy). Random photomicrographs are presented (400× magnification). In the photo micrograph of (**d**,**g**) * represents 400× magnification, same region in enlarged magnification 1000× (**). ns—no significant. The yellow arrow represents the positive cells.

**Figure 6 cells-13-00295-f006:**
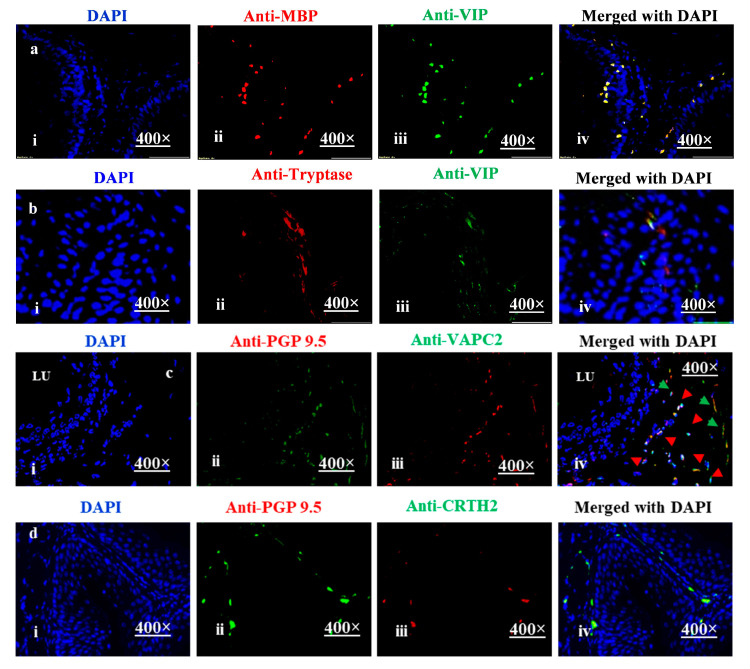
**Accumulation of eosinophils and mast cells in muscular mucosa following the induction of experimental EoE**. Eosinophils and mast cells in mouse esophageal tissue sections were detected by anti-MBP immunostaining and chloroacetate staining, respectively. Both eosinophils and mast cells were detected in each segment of mouse esophagus following the induction of experimental EoE. Eosinophil accumulation was observed in epithelial mucosa, lamina propria, and muscular mucosa (**a**(**i**–**iv**)). The accumulation of mast cells was observed mainly in lamina propria and muscular mucosa (**b**(**i**–**iv**)). Double immunofluorescence of anti-PGP 9.5 in muscular mucosa, a nerve cell marker, and anti-VAPC2, anti-CRTH2 receptors were observed in nearby cells (**c**(**i**–**iv**),**d**(**i**–**iv**)). Randomly selected representative photomicrographs of patient biopsies are shown at 100× and 400× original magnification (*n* = 5, 3–4 field/hpf field/tissue section). The green arrow indicates PGP 9.5 ^+^ nerve cells, red arrow indicates mast cell receptor VAPC2.

**Figure 7 cells-13-00295-f007:**
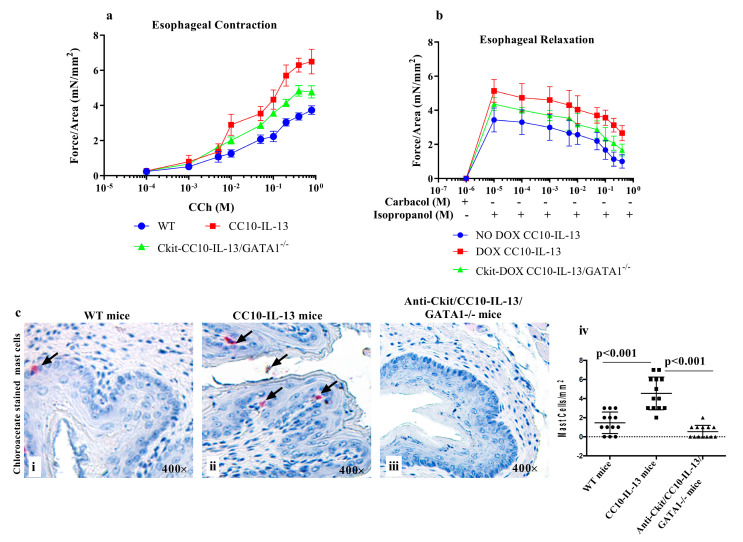
**Esophageal motility impairment is independent of eosinophilic inflammation**. Concentration-dependent, carbachol-induced contraction of esophageal longitudinal muscle and isoproterenol-induced relaxation of carbachol-precontracted (10^5^ M) esophageal longitudinal muscle contraction were measured in WT, rtTA-CC10-IL-13, and anti-C-kit-neutralized ΔdblGATA/rtTA-CC10-IL-13 mice by adding increasing concentrations of carbachol or isoproterenol (**a**,**b**). A representative photomicrograph of anti-C-kit-neutralized ΔdblGATA/ rtTA-CC10-IL-13 mice shows significantly reduced mast cell accumulation in the esophagus compared to that in rtTA-CC10-IL-13 mice. Data are presented from 4 mice/group after analyzing 12–13 400× fields/samples (**c**(**i**–**iv**)). The arrow marks indicate the presence of mast cells.

**Figure 8 cells-13-00295-f008:**
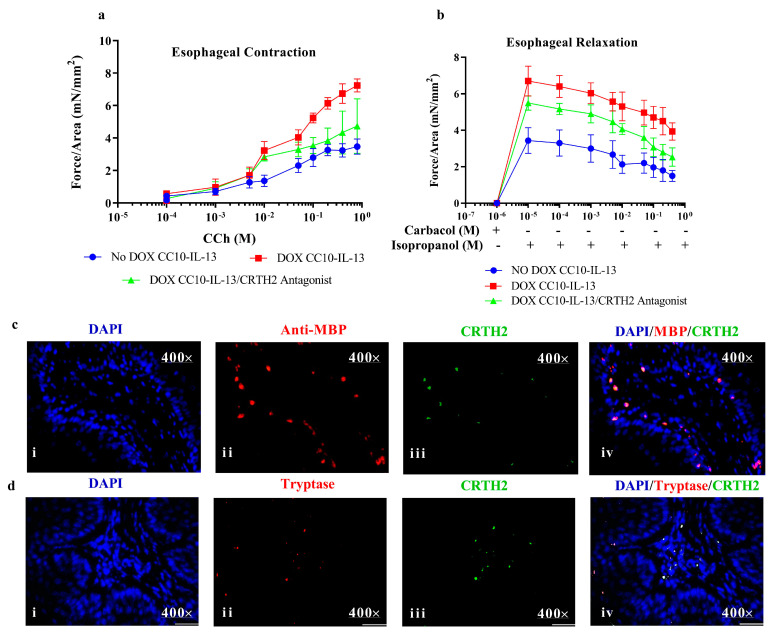
**CRTH2 antagonist treatment improves impaired longitudinal esophageal dysfunction in a DOX-regulated CC10-IL-13-overexpressed mouse model of EoE**. Esophageal muscle contraction and relaxation in rtTA-CC10-IL-13 mice is impaired compared with no-DOX rtTA-CC10-IL-13 mice. Comparable esophageal motility impairment is shown between rtTA-CC10-IL-13 mice treated with rtTA-CC10-IL-13 and CRTH2 antagonist (**a**,**b**). Immunofluorescence analysis of eosinophils and CRTH2 (**c**(**i**–**iv**)) and mast cells and CRTH2 (**d**(**i**–**iv**)) in CRTH2 antagonist-treated DOX-exposed rtTA-CC10-IL-13 mice (sections for n = 6 studied; photographs taken randomly). Data are expressed as means ± SD, *n* = 6 mice/group, *p* < 0.01.

## Data Availability

The data presented in this study are available in the article.

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
