# Peer review of "Vasoactive Intestinal Peptide Receptor, CRTH2, Antagonist Treatment Improves Eosinophil and Mast Cell-Mediated Esophageal Remodeling and Motility Dysfunction in Eosinophilic Esophagitis"

_cells, 2024, doi:10.3390/cells13040295_

Round 1
Reviewer 1 Report (Previous Reviewer 2)
Comments and Suggestions for Authors
This submission described the effect of VIP receptor CRTH2 antagonist treatment on eosinophils and mast cells in EoE. The workload was adequate and the logic was clear. I recommend a major revision on the following points:
1. The format was messy. The authors should pay attention to the line space, standard unit, tense use, and font style. The language use should also be improved.
2. In the introduction, the authors should add more detailed demonstration on the significance of this work. I also recommend to add the current therapeutical approaches for treating EoE. The advantages of selecting CRTH2 antagonist treatment should also be added.
3. For the CRTH2 and VAPC2 receptor expression on tissue-accumulated human eosinophils and mast cells, what was the logic sequence. First the induction of the expression than the migration, or first the migration to a clos position to induce the expression?
4. The side effect of doxycycline should be tested or claimed.
Comments on the Quality of English LanguageModerate editing required.
Author Response
Dear Reviewer,
Please see the attachment of the authors response for the reviewer comments.

Reviewer 2 Report (New Reviewer)
Comments and Suggestions for Authors
Comments:
1) The authors need to fix the typos all over the manuscript. The VIP receptor VPAC1/VAPC1 is not clear.
2) Analysis of VIP and VIP Receptors (VACP1, VAPC2 and CRTH2) on Eosinophils and Mast Cells and the VIP Receptor’s Role in the Migration of both Inflammatory Cells. Could the authors demonstrate or provide experimental evidence to show the interaction between CRTH2/VAPC2 and VIP is essential for eosinophil motility.
3) Fig 2, the correlation between Eos vs CRTH2, both the P value and the r values doesn’t match.
4) Fig 3, can the authors show representative H&E of an human esophageal biopsy.
5) In the supplementary figure 2A, the H&E was from human esophageal biopsy, but the muscularis mucosa inset view as well as the nerve cells and ganglion from colon biopsies what are the authors trying to interpret here, because these two interpretation are from different gastrointestinal origin.
6) In 3.5 and Fig 4, the authors were trying to discuss the IF image that shows all segments of esophageal biopsy detects the eosinophils using Anti-MBP and mast cells using Anti-tryptase, and the nerve cells using Anti-PGP 9.5 in epithelial mucosa. Could the authors provide more info on anti-CRTH2 and anti-VAPC2 IF images in addition to the Anti-MBP, Anti-VIP, Anti-PGP 9.5 and Anti-tryptase to show that the accumulation of eosinophils and mast cells near the nerve cells are more specific and also provide some details regarding the quantification on how many fields were chosen to interpret their results.
7) In Fig 5 and 6, since the authors mentioned evidence for accumulation of eosinophils and mast cells adjacent to nerve cells in the different segments of esophageal biopsy, Could the authors provide Anti-PGP 9.5, Anti-VIP as well. Additionally, could they provide details on the VIP receptors for the same to show the specificity of their results. Also could the authors provide details on the quantification of eosinophils and mast cells expressing VIP receptors.
Author Response
Dear Reviewer
Please see the attachment for the author's response to the reviewer's comments.

Round 2
Reviewer 1 Report (Previous Reviewer 2)
Comments and Suggestions for Authors
The authors have improved the quality of the submission. I recommend the acceptance.
This manuscript is a resubmission of an earlier submission. The following is a list of the peer review reports and author responses from that submission.
Round 1
Reviewer 1 Report
Comments and Suggestions for Authors
Yadavalli and colleagues investigate the role of vasoactive intestinal peptide (VIP) on granulocytes like eosinophils and basophils in the context of eosinophilic esophagitis (EoE). Using both mouse and human models and tissues, the authors aim to phenotype and mechanistically pinpoint the effect of VIP on the pathology. The approach used is both interesting and adequate, with human patient data and genetically modified mouse strains. The subject is relevant to the field and definitely worth investigating. At the moment, the manuscript is plagued by poor writing and inaccuracies in the Methods and Results section, including the figures. First, the authors are encouraged to rewrite the manuscript with the help of an academic native English speaker, as the current state of the paper is riddled with typos and incorrect English. The figure legends require accurate statistics, mention of repeated experiments and significance should be clear in the figure itself. Second, the figures are not elaborate enough to support the results. Thirdly, the mouse experiments miss critical controls, like verification of mast cell depletion after 2B8 administration, expression of VIP and the receptors on mouse eosinophils and mast cells by flow cytometry, Eos/Mast cell deficient mice (GATA1-/-) only in the motility experiments etc. While I do see the potential, the current form of the manuscript has too many flaws to continue the review process. I encourage to consider adjusting the manuscript and resubmit the manuscript at a later date.
Abstract: Several preventable typos make reading difficult, for example “Quantitative PCR analyses showed that VIP levels its receptor CRTH2 were increased ~ 5 and ~6- 21 fold, respectively…”. And “Interestingly, the levels of CRTH2 mRNA corelate with the number of eosinophils as well the levels of VIP and chemoattract to nerve cell-derived VIP in human EoE (p<0.05).”
Introduction:
Rewrite 40-42, unclear sentence
Rewrite 45-47, unclear sentence
Rewrite 50-52, unreadable sentence
Rewrite 61-64, unclear sentence
Materials and methods: 1mg of anti-c-kit antibody or 5 times is an enormous amount of antibody. The 2B8 clone has not been reported to deplete mast cells, it is the ACK2 clone that does that. Report on the exact depletion of immune subsets in this treatment regimen or provide a study describing this. Describe the clones and fluorochromes used in the flow cytometry experiments, include the manufacturer per antibody. How many cells were cultured in how much IL-3 cytokine obtained where?
Rewrite line 187, this cannot be correct: “intranasal injection of 100 μg of Aspergillus fumigatus (Greer Laboratories, Lenoir, NC) in 50 μg of normal saline or 50 g of normal saline”
2.11 Statistical analysis does not make any sense. The authors mention a t-test between two groups, however, their first figure contains multiple groups for which no statistical test specifics are given (and cannot be analyzed using a t-test). And statistical significance calculated using a Spearman correlation test? How?
Results:
Figure 1: Be consistent when labeling the X axes: either choose “VIP” or “VIP expression”. Statistics should be described in the figure legend, not in the text line 280-282.
Figure 2: This whole figure in inaccurately described by the figure legend. 2a is serum VIP protein, but described as mRNA expression in the legend? Statistics are lacking? Figure 2e and 2f are not correctly mentioned in the text. Specifics on statistics should be in the Figure legend, not the main Results text. Figure 2c VAPC-1 mRNA is described as not significant, which I find hard to believe. Please provide the raw data points for this figure so I can perform statistics.
Figure 3: stainings are represented or how many fields of view? All patients? Where in the tissue is the image taken? Please describe in the text.
Line 341-344: Unreadable sentence, please rewrite. Line 348: please rewrite.
Line 339-353 mentions Supplementary Figure 3, which is not correct. Also, the supplementary figure 2 containing the overview staining should be in the main figure 3 or 4, it is part of the main evidence put forward by the authors.
Figure 4: Please be consistent when labeling the antibodies used (VIP or anti-VIP). Include statistics in the bar graph. How did you define the groups “adjacent nerve cells” and “away nerve cells”? This should be mentioned in the text and figure legend.
Line 377: The accumulation of mast cells (tryptase+) around VIP-expressing cells is not convincing, since there is a direct overlap of tryptase and VIP staining. Can mast cells produce VIP in the epithelial mucosa?
Figure 5: Include statistics in the bar graph. Please do not just rotate the H&E figure making the text upside-down.
Line 433: What is DOX? Write out first, then abbreviate.
Figure 6: Eosinophils produce VIP? Perfect overlay off MBP and VIP signal?
Figure 7/8: Expression of VIP and its receptors should be investigated on mouse eosinophils and mast cells in this model to make any parallel claim to these cells being responsible for esophageal dysunction.
Discussion:
The authors should not overstate their findings. While they should eosinophils accumulation, it is not clear this is due to VIP or eotaxin-3. Consider the hypothesis that eosinophils infiltrate randomly, this perfectly fits the data. Eosinophils accumulate regardless of the location of neurons.
Comments on the Quality of English LanguageThe English isn't great, please consult a native English academic to rewrite the sentences and remove typos. At the moment, the message is often unclear and this severely hampers the scientific communication.
Reviewer 2 Report
Comments and Suggestions for Authors
This submission described the correlation between CRTH2 and eosinophils as well as mast cells-mediated esophageal remodeling. The mechanic investigation was detailed and step-by-step. I recommend a major revision on the following points:
1. The VIP receptor CRTH2 antagonist should be detailed. The structures, sequences, specificity, and mechanisms should be explained. The authors should confirm that the effect was particularly associated to CRTH2.
2. The expression of CRTH2 receptor on the tissue accumulated eosinophils and mast cells in the biopsies of human EoE should be explained in detail together with the concentration of the substrates, because both the receptor level and the signal level were important.
3. The confocal images showed some overexposure. If possible, the authors should adjust the contrast and brightness.
4. In the discussion section, I recommend the authors find some references on the knock-out of the key receptors.
5. The language use should be improved.
Comments on the Quality of English LanguageMinor editing required.